# Self-Amplifying RNA: A Second Revolution of mRNA Vaccines against COVID-19

**DOI:** 10.3390/vaccines12030318

**Published:** 2024-03-17

**Authors:** Noelia Silva-Pilipich, Uxue Beloki, Laura Salaberry, Cristian Smerdou

**Affiliations:** 1Division of DNA and RNA Medicine, Cima Universidad de Navarra, 31008 Pamplona, Spain; ubeloki@alumni.unav.es; 2Instituto de Investigación Sanitaria de Navarra (IdISNA) and CCUN, 31008 Pamplona, Spain; 3Facultad de Ingeniería, Universidad ORT Uruguay, Montevideo 11100, Uruguay; salabelau@gmail.com; 4Nanogrow Biotech, Montevideo 11500, Uruguay

**Keywords:** COVID-19 vaccines, mRNA vaccine, self-amplifying RNA, alphavirus, clinical trials

## Abstract

SARS-CoV-2 virus, the causative agent of COVID-19, has produced the largest pandemic in the 21st century, becoming a very serious health problem worldwide. To prevent COVID-19 disease and infection, a large number of vaccines have been developed and approved in record time, including new vaccines based on mRNA encapsulated in lipid nanoparticles. While mRNA-based vaccines have proven to be safe and effective, they are more expensive to produce compared to conventional vaccines. A special type of mRNA vaccine is based on self-amplifying RNA (saRNA) derived from the genome of RNA viruses, mainly alphaviruses. These saRNAs encode a viral replicase in addition to the antigen, usually the SARS-CoV-2 spike protein. The replicase can amplify the saRNA in transfected cells, potentially reducing the amount of RNA needed for vaccination and promoting interferon I responses that can enhance adaptive immunity. Preclinical studies with saRNA-based COVID-19 vaccines in diverse animal models have demonstrated the induction of robust protective immune responses, similar to conventional mRNA but at lower doses. Initial clinical trials have confirmed the safety and immunogenicity of saRNA-based vaccines in individuals that had previously received authorized COVID-19 vaccines. These findings have led to the recent approval of two of these vaccines by the national drug agencies of India and Japan, underscoring the promising potential of this technology.

## 1. Introduction

### 1.1. COVID-19 and SARS-CoV-2

COVID-19 is a respiratory disease caused by SARS-CoV-2 virus, a new type of human Beta-coronavirus that appeared at the end of 2019 in Wuhan (China) causing a global pandemic still present in many areas of the world. To date, this pandemic has produced more than seven million deaths and 774 million cases worldwide (data from February 2024) [1]. COVID-19 causes flu-like symptoms, such as fever, coughing, sore throat, fatigue, and muscle aches. In some cases, the infection can lead to severe pneumonia and acute respiratory distress syndrome (ARDS), which is the main cause of death [2]. Since its first appearance in 2019, the virus has evolved very quickly, giving rise to variants and subvariants with different abilities to propagate and cause disease. Currently, the most prominent variants of concern (VoCs) present in the population derive from the so-called omicron variant, which is able to propagate extremely efficiently through aerosols, although it seems to be less pathogenic compared to earlier VoCs, such as the Wuhan, Alpha, or Delta variants [3].

SARS-CoV-2 is an enveloped virus containing a positive-sense single strand RNA genome of about 30 kb in length. The viral particle contains four different structural proteins: S (spike), N (nucleocapsid), M (membrane), and E (envelope). The S protein is a 180–200 kDa glycoprotein of 1273 amino acids with two main domains (S1 and S2), which is assembled into trimers on the surface of the virion. S is usually in a stable prefusion conformation until it binds to its receptor, the human angiotensin-converting enzyme 2 (hACE2) receptor on the surface of target cells. For this recognition, the S protein has a specific domain called the receptor binding domain (RBD). hACE2 is very abundant in respiratory endothelial cells, serving as the main gate of entry for the virus. Many studies have shown that antibodies against SARS-CoV-2 S protein, and especially those directed against the RBD, are neutralizing since they can efficiently block the attachment of the virus to target cells [4]. For this reason, most of the vaccines developed against COVID-19 are based on formulations able to induce anti-S neutralizing antibodies (nAbs), as will be discussed in the next section.

### 1.2. Approved COVID-19 Vaccines

The development of efficient COVID-19 vaccines just a few months after the identification of SARS-CoV-2 is considered one of the most spectacular achievements in the history of vaccination. Several factors contributed to this superfast-track process, among them the huge and broad accumulated knowledge on coronaviruses, the rapid sequencing of the SARS-CoV-2 genome, and the availability of state-of-the-art technologies to produce vaccines based on either messenger RNAs (mRNAs) or recombinant viral vectors. An additional factor that was crucial for the rapid approval of these vaccines was the rapid expansion of SARS-CoV-2 in the world population, affecting millions of individuals in a very short time. This created a global emergency that accelerated the regulatory procedures necessary for the approval of new vaccines by national medicine agencies. In addition, having so many people simultaneously exposed to the virus created optimal conditions for conducting large-scale testing of the new vaccines, allowing the execution of phase III clinical trials with cohorts composed of tens of thousands of individuals, an exceptional circumstance in vaccine testing. Although nowadays, there are more than fifty different COVID-19 vaccines approved worldwide (Table 1), they can be classified into four main categories: inactivated virus vaccines, subunit vaccines based on recombinant viral proteins, non-replicating viral vector vaccines, and nucleic acid vaccines, mostly based on mRNA (although a DNA vaccine has also been recently approved in India) [5]. mRNA and viral vector vaccines were the first to be authorized by both the Food and Drug Administration (FDA) and the European Medicine Agency (EMA). The approved viral vector vaccines are based on a recombinant adenovirus expressing the S protein and include the Oxford–AstraZeneca (AZD1222, approved only by EMA) and the Janssen COVID-19 (approved by both EMA and FDA) vaccines [6,7]. In both vaccines, the vector contains the gene coding for the S protein, but they use different adenovirus serotypes: chimpanzee adenovirus (ChAd) Ox1 and human adenovirus 26, for Oxford-AstraZeneca and Janssen, respectively. Other adenovirus-based vaccines that have been approved and extensively used outside Europe and the USA include the Sputnik V and Convidecia vaccines, developed in Russia and China, respectively [8]. 

The development of mRNA vaccines was possible thanks to two previous innovations involving the mRNA itself and its delivery vehicle. In the first case, Katalin Karikó and Drew Weissman, who received the 2023 Nobel Prize in Physiology or Medicine [9], demonstrated that the introduction of nucleoside modifications into mRNA could reduce innate responses that were responsible for its rapid degradation in vivo. When mRNA is introduced in vivo it can signal through different toll like receptors (like TLR3, TLR7, and TLR8), but the incorporation of modified nucleosides, like pseudouridine, ablates their activity, expanding the half-life of RNA and prolonging its expression [10]. 

The entry of nucleic acids into cells is a very inefficient process due to the large size of these molecules and to their hydrophilic nature, which does not allow them to easily cross the hydrophobic lipid bilayer that constitutes the cell membrane. To facilitate entry of mRNA into cells, it can be coupled to cationic lipids or polymers that can both neutralize the numerous negative charges of mRNA and provide it with the necessary hydrophobicity to pass through membranes. Although these formulations have been used for a long time, their initial compositions were not optimal and were rather inefficient for in vivo delivery. However, the recent development of lipid nanoparticles (LNPs) with optimized formulations has revolutionized this field. LNPs used for COVID-19 vaccines are usually composed of four components that include ionizable lipids, phospholipids, cholesterol, and PEGylated lipids (for a review on LNPs see [11]). Two mRNA-based COVID-19 vaccines were rapidly approved by both the FDA and EMA, which were developed by Moderna (mRNA-1273) [12] and Pfizer-BioNtech (BNT162b2) [13]. Both vaccines utilize LNPs containing mRNA of approximately 4 kb coding for a version of the S protein carrying two mutations (K986P and V987P) designed to stabilize the pre-fusion conformation of this protein. Both mRNA vaccines have shown a high degree of safety and a very high efficacy providing >90% protection against infection and drastically reducing the hospitalization rate in people receiving at least two doses [14]. Despite the good performance of mRNA vaccines, they also present some shortcomings, such as their high cost and the fact that their storage and transportation require very low temperatures, rendering them inaccessible for many countries.

**Table 1 vaccines-12-00318-t001:** Most relevant approved COVID-19 vaccines worldwide.

Vaccine Name	Company	Country of Origin	Platform ^a^	Antigen ^b^	First Approval	Reference
mRNA-1273	Moderna	USA	mRNA LNPs	Pre-fusion S	2020	[12]
Moderna 2023–2024 Formula	Moderna	USA	mRNA LNPs	Pre-fusion S (Omicron XBB.1.5)	2023	[15]
BNT162b2	Pfizer/ BioNTech	Germany/USA	mRNA LNPs	Pre-fusion S	2020	[13]
Pfizer 2023–2024 Formula	Pfizer/ BioNTech	Germany/USA	mRNA LNPs	Pre-fusion S (Omicron XBB.1.5)	2023	[15]
AZD1222	AstraZeneca/Oxford	UK	ChAdOX1	Native S	2021	[6]
Ad26COVS1	Janssen	Holland	Ad26	Pre-fusion S	2021	[7]
Gam-COVID-Vax Sputnik V	Gamaleya Research Institute	Russia	Ad26/Ad5	Native S	2020	[16]
NVX-CoV2373	Novavax	USA	Protein-based	Pre-fusion S	2021	[17]
CoronaVac	SinovacBiotech	China	Inactivated virus	Whole inactivated virus	2021	[18]
Convidecia™ Ad5-nCoV	CanSino	China	Ad5	Native S	2021	[19]
ZyCov-D	Zydus Cadila/ India’s Depart. Biotechnology	India	Plasmid DNA	Native S	2021	[20]

^a^ LNPs, lipid nanoparticles; Ad, human adenovirus; ChAd, chimpanzee adenovirus. ^b^ Unless otherwise indicated, the S protein (or whole inactivated virus) corresponds to the original Wuhan SARS-CoV-2 variant.

### 1.3. Self-Amplifying RNA

A different type of mRNA with the ability to self-replicate has also been used to develop COVID-19 vaccines. Self-amplifying RNA (saRNA)-based vaccines have the advantage of requiring much lower doses than conventional mRNA vaccines to obtain similar immune responses. Although their development has been slower compared to mRNA vaccines, saRNA COVID-19 vaccines have already been tested in numerous preclinical and clinical trials, as will be discussed in this review, with two vaccines recently approved in India and Japan [21]. saRNA vaccines are derived from the genomes of RNA viruses, such as alphaviruses, flaviviruses, and rhabdoviruses, among others. Since the vast majority of saRNA vaccines for COVID-19 that have entered clinical trials thus far are based on alphavirus vectors, this review will focus on this type of vaccines.

Alphaviruses are enveloped viruses containing a positive-strand RNA genome of approximately 12 kb. This genome contains two open reading frames (ORFs), one coding for the viral replicase (Rep) and a second one coding for a polyprotein that is processed to produce the viral structural proteins: capsid, p62 (precursor of E2), 6K, and E1. Once the virus infects a cell, Rep is translated from the genomic RNA being able to synthetize a complementary negative-strand RNA. This RNA is used as a template to amplify the viral genomic RNA. For these processes, Rep recognizes 5′ and 3′ untranslated regions (UTRs) containing secondary structures. Rep is also able to make a smaller subgenomic mRNA from which the viral structural polyprotein will be translated at very high levels. Alphavirus vector RNAs are usually generated by substituting the ORF coding for the structural proteins by the gene of interest [22], as depicted in Figure 1. This type of saRNA can be transcribed in vitro from a plasmid containing its sequence as it is routinely done to produce conventional mRNAs. Alphavirus RNA vectors can be employed directly as RNA or can be packaged into viral particles by providing the viral structural proteins in *trans,* using one or two helper RNAs that are co-transfected together with the vector RNA into packaging cells [22]. In addition, alphavirus saRNA vectors can be delivered as DNA by a DNA/RNA layered system in which the sequence of the RNA vector is placed downstream of a eukaryotic promoter [23]. saRNA vectors have some intrinsic properties that make them particularly interesting as vaccines: (i) they are able to express very high amounts of antigen [24], (ii) their expression is transient due to the induction of apoptosis in transfected cells after two-three days [25], (iii) they induce type I interferon (IFN-I) responses that can enhance adaptive immunity [26,27], and (iv) they are not integrative, making them quite safe. 

Alphavirus vectors have been developed from three main viruses: Semliki Forest virus (SFV), Venezuelan Equine Encephalitis virus (VEEV), and Sindbis virus (SIN). Vectors derived from these viruses have been used in numerous preclinical studies for both vaccination [28] and cancer gene therapy [29]. Before their use as COVID-19 vaccines, some alphavirus vectors had been tested in clinical trials to vaccinate against cancer or viral antigens, although they were always used as viral particles [30]. The possibility to deliver these vectors directly as RNA has only been clinically explored after the success of mRNA vaccines for COVID-19, as will be discussed in this review.

## 2. Delivery Strategies for saRNA Vaccines against SARS-CoV-2

A variety of animal models, from mice to non-human primates (NHPs), have been used to evaluate the efficacy, biodistribution, and safety of SARS-CoV-2 vaccines. This has allowed the rapid translation into clinical trials of several SARS-CoV-2 vaccines, including the ones based on mRNA and, more recently, saRNA. Since mice and rats are naturally resistant to SARS-CoV-2 infection, they are not suitable to test the ability of vaccines to prevent viral infection, replication, and transmission. However, some animals have been described to be naturally susceptible to SARS-CoV-2 infection, such as hamsters, ferrets, cats, and monkeys, although they usually do not show severe symptoms of disease or mortality as humans. In this regard, the development of mouse models susceptible to SARS-CoV-2 pathogenesis has helped accelerate the development of vaccines and therapeutics for COVID-19. These models include genetically modified mice expressing hACE2 or mice transduced with adeno-associated virus (AAV) or adenoviral vectors encoding hACE2 [31].

One of the most challenging aspects of nucleic acid therapy is to achieve an efficient delivery of these molecules into cells. In this regard, several non-viral strategies have been developed including LNPs and other less conventional alternatives. Currently, LNPs are the most advanced technology, as evidenced by the approval of Onpattro in 2018, a small interfering RNA (siRNA)-based therapy delivered by LNPs for the treatment of transthyretin amyloidosis [32]. This approval was a fundamental piece in the unprecedented rate at which SARS-CoV-2 mRNA vaccines were designed, developed, and produced at high scale. However, saRNA molecules differ considerably from mRNA, not only in terms of size (saRNA is three to four times longer than mRNA), but also on their ability to stimulate the immune system due to RNA self-replication. Different delivery vehicles, administration routes, and dosing regimens have been evaluated in preclinical studies involving SARS-CoV-2 saRNA vaccines (Figure 2 and Table 2). These studies have overall demonstrated that this type of vaccine is able to induce potent nAb responses and T cell immunity with a good safety profile, which has paved the way for their clinical evaluation.

### 2.1. LNPs for the Delivery of saRNA

The use of LNPs for the delivery saRNA vaccines has been evaluated extensively since the first attempt, published more than a decade ago [33]. Several articles have illustrated the importance of optimizing LNPs for saRNA-based vaccines, including the formulation, administration routes, storage conditions, and even the RNA itself [34,35,36,37]. 

Most of the vaccines against SARS-CoV-2 are based on the S protein, which is the major target for nAbs. Many vaccine candidates, including some based on saRNA, have been developed from a mutated version of the S protein, which is stabilized in the prefusion conformation exposing the RBD, a hotspot for nAbs. In the work by Maruggi et al., the full-length S protein of the Wuhan-Hu-1 isolate, stabilized in the prefusion conformation, was cloned into a VEEV-based replicon and delivered using LNPs [38]. nAb titers were induced in mice in a dose-dependent manner after one dose and boosted 30-fold by a second immunization. Sera from immunized mice were also able to neutralize the Alpha (B.1.1.7), Beta (B.1.351), and Delta (B.1.617.2) VoCs, although less efficiently compared to the Wuhan-Hu-1 strain. This cross-reactivity could be due to the fact that VoCs share some common spike epitopes. Vaccine cross-reactivity seems to be lower with variants having a high number of changes, such as Omicron, which contains 37 amino acid substitutions in the spike protein compared to the original Wuhan strain, 15 of which are in the RBD. But even in this case, individuals that received vaccines based on the Wuhan strain such as BNT162b2 showed residual neutralization against Omicron [39], suggesting that at least some neutralizing antibodies could be present, although they might not be very abundant or might not have an optimal activity. The induction of broadly neutralizing antibodies could be key to generate a potent vaccine against COVID-19.

A similar strategy based on a VEEV replicon encoding the prefusion stabilized SARS-CoV-2 S protein and encapsulated in LNPs was tested in mice, with doses ranging from 0.01 to 10 µg, in a prime/boost setting [40]. This strategy also induced high and dose-dependent specific nAbs as well as a Th1-biased response. In a subsequent work, the authors tested this strategy in a contact transmission model in Golden Syrian hamsters, in which infected animals were co-housed with animals that had received a prime-boost vaccination. Vaccination was able to protect animals from severe disease using two different strains, a historic isolate from the summer of 2020 (B.1.238) and Alpha VoC (B.1.1.7). Although animals were not protected against infection, weight loss and viral shedding were reduced compared to sentinel animals vaccinated with an irrelevant saRNA. In this model, routes of potential infection, as well as viral doses, could be considered more natural compared to conventional direct intranasal (IN) inoculation [41].

At least two doses of the currently approved mRNA vaccines are needed to elicit robust immunity, which increases the costs and makes it harder to reach full protection in a big fraction of the population. One of the main advantages of using saRNA as a platform for vaccination is that it could potentially allow a lower dose and/or avoid the need of a boost. In the work by de Alwis et al., a single dose of saRNA based on the VEEV replicon, encoding unmodified S protein and encapsulated in LNPs (named LUNAR-COV19), was compared to its mRNA counterpart in mice. At all tested doses, LUNAR-COV19 generated significantly higher S-specific IgG titers with Th1-skewed IgG subclasses, and higher SARS-CoV-2-specific cellular responses compared to a dose-matched conventional mRNA control. In addition, hACE2 transgenic mice were protected from SARS-CoV-2 infection and mortality after a single dose of LUNAR-COV19 with both 2 and 10 µg doses. These results support the enhanced ability of saRNA vaccines to induce robust immunity after a single and lower dose compared to mRNA, which could decrease the pressure on the manufacturing process. saRNA was well-tolerated in all animals except in the case of the highest dose, where body weight loss and other clinical score changes were observed, underlying the importance of dose optimization [42].

### 2.2. New Trends in Lipid-Based Formulations for saRNA Delivery

In addition to LNPs, a number of alternative lipid-based delivery systems for saRNA vaccines have been described. Despite the prevailing notion that encapsulation of RNA is needed for its protection against degradation, some studies have shown that RNA can adhere to the surface of lipid carriers and maintain its integrity even in the presence of RNases [43]. This concept unlocks opportunities for designing easier-to-manufacture lipid formulations that could potentially be stockpiled for pandemic preparedness in the absence of the target RNA. Many of these alternatives to LNPs offer additional advantages, including higher thermostability, simpler and more scalable manufacturing processes, and formulations free from proprietary ionizable lipids, reducing the risk of supply shortages. Combining these advantages with the potency and self-adjuvating nature of saRNA holds promise for “next-generation” SARS-CoV-2 RNA vaccines. 

One example is the nanostructured lipid carrier (NLC), which is based on nanoparticles with a hybrid core containing a liquid oil phase, such as squalene, and a solid phase lipid composed of a saturated triglyceride. They also contain nonionic surfactants, the hydrophobic sorbitan ester (Span), the hydrophilic ethoxylated sorbitan ester (Tween), and the cationic lipid DOTAP (1,2-Dioleoyl-3-trimethylammonium propane), which interacts with the RNA on the surface of the nanoparticle. These surfactants are not only critical for the long-term colloidal stability of NLC, but also influence the efficacy of protection against RNase degradation and the immunogenicity of the vaccines. An extensive optimization of the NLC formulation for the delivery of a saRNA encoding Zika virus antigens was performed by Erasmus and colleagues [44]. A more comprehensive study regarding the stability of this platform demonstrated that the NLC formulation is stable for at least one year at 4 °C without RNA complexation. In addition, NLC complexed with RNA can be lyophilized and stored refrigerated or at room temperature for at least 21 or 8 months, respectively [45]. The feasibility of using NLC/saRNA in the context of SARS-CoV-2 vaccination was evaluated using a VEEV-based replicon encoding the S protein [46]. Strong antibody responses to the S protein were detected in mice immunized intramuscularly (IM) using different doses. Enhancement of cross-variant nAbs was achieved by using an optimized sequence of the S protein stabilized in the prefusion conformation and in which the furin cleavage site was ablated (AAHI-SC2). Robust Th1-biased T cell responses and nAbs against Wuhan, Alpha (B.1.1.7), and to a lower degree Beta (B.1.351) and Delta (B.1.617.2) VoCs were achieved in mice vaccinated with AAHI-SC2. In addition, AAHI-SC2 exhibited long-term thermostability after lyophilization, and could be stored at room temperature for at least six months without loss of RNA integrity or vaccine efficacy in mice [46]. 

Erasmus and colleagues also developed a platform called LION (for Lipid InOrganic Nanoparticle), an emulsion aimed at enhancing vaccine stability, intracellular delivery, and immunogenicity [47]. LION is a highly stable cationic squalene emulsion with 15 nm superparamagnetic iron oxide (Fe_3_O_4_) nanoparticles embedded in a hydrophobic oil phase and contains DOTAP, which enables electrostatic association with saRNA molecules at the nanoparticle’s surface, avoiding the need for an encapsulation process. Manufacturing and stockpiling LION independently of the saRNA component is possible since it is stable at 4 °C and it can be later complexed with the appropriate saRNA in a simple mixing step. In addition, the vaccine was shown to be stable for at least one week at 25 °C after saRNA complexation. In mice, a LION-saRNA vaccine based on VEEV induced a Th1-biased antibody response and S-specific T cells at doses of 10, 1, and 0.1 μg of RNA. This vaccine was also tested in pigtail macaques using a single dose of 250 μg or two doses of 50 μg with a 4-week interval. Neither adverse reactions at the injection sites nor abnormalities in body weight or temperature were observed in the animals. In terms of efficacy, a rather modest S-specific T cell response was achieved, although anti-S antibody titers showed a prolonged increase until day 42 and plateaued until at least day 70 post-vaccination, probably due to the sustained antigen expression mediated by the saRNA [47]. Given this time-course and the fact that both doses generated similar responses, in a subsequent study the authors evaluated the possibility of reducing the dose and spacing the boost [48]. Interestingly, they observed that increasing the interval between the prime and boost to 20 weeks generated a more robust and durable nAb response in pigtail macaques, using a lower dose of 25 μg. After challenge with SARS-CoV-2 (WA-1 strain), vaccinated animals showed lower viral loads and accelerated viral clearance in the nasal and lung compartments compared to unvaccinated controls, as well as protection against clinical disease and lung pathology [48]. In addition, this vaccine was evaluated in a proof-of-concept study using simian immunodeficiency virus (SIV)-infected pigtail macaques. A single dose of the vaccine was able to trigger humoral responses against the S protein in some animals, with a slightly better induction of nAbs in those receiving the highest dose (25 µg) compared to the lowest dose (5 µg) [49]. Although cellular responses were modest, this pilot study suggests that saRNA vaccines could be employed successfully in HIV-infected and other immunocompromised individuals, a particularly vulnerable population where the efficacy of approved SARS-CoV-2 mRNA vaccines has been reported to be lower [50,51,52].

The technology involving LION has several advantages compared to the currently approved mRNA vaccines based on LNPs. The increase in the stability of the formulation makes distribution and storage easier, particularly in countries where infrastructures are unprepared to maintain the cold chain. The saRNA-LION vaccine has been evaluated in clinical trials in India, leading to its recent approval in this country under the name of HDT-301. This represents a significant milestone since it is the first saRNA vaccine ever approved for human use. Another advantage of this platform is the possibility of rapid adaptation to emergent VoCs or even to future pandemics caused by different pathogens, since the formulation can be easily complexed with other saRNA molecules. In the work by Hawman and colleagues, the authors updated their previous LION vaccine [47] by changing the S protein sequence to match those of B.1, B.1.1.7, and B.1.351 VoCs, and evaluated the efficacy of these new vaccines against homologous or heterologous SARS-CoV-2 challenge in mice and hamsters [53]. All the vaccines were able to induce nAb responses against matched or mismatched VoCs, although the B.1.351 variant showed higher resistance to nAbs induced by the other vaccines. Vaccination with 20 μg of LION/saRNA with the same prime/boost schedule in Golden Syrian hamsters also induced homologous and heterologous nAbs. Four weeks after the boost, hamsters were challenged with the A.1, B.1.351, or B.1.1.7 VoCs. Vaccination with any of the saRNAs significantly reduced viral shedding in the upper airway, viral burden in lung, and lung pathology, in both homologous and heterologous SARS-CoV2 challenges. However, superior protection and a more rapid clearance against homologous VoCs challenge was observed compared to heterologous challenge. These results indicate that the LION/saRNA vaccine is able to induce broadly protective immunity against multiple SARS-CoV-2 VoCs [53].

One strategy to increase cross-neutralizing activity is the design of multivalent vaccines. This might require increasing the total amount of saRNA per dose, which could compromise the safety of the vaccine, since some clinical studies using saRNA delivered by LNPs have shown dose-limiting reactogenicity [54,55]. In a biodistribution study, saRNA delivered by LNPs was shown to disseminate to distant organs, in contrast to LION-formulated saRNA, which localized at the injected muscle and draining lymph nodes (dLNs). As a result, mice vaccinated with LION/saRNA showed activation of innate immune responses restricted to the injection site, without systemic inflammation or significant weight loss [56]. These results suggest that, in the context of saRNA, alternative delivery platforms distinct from LNPs such as LION, may be more suitable for eliciting optimal immunity with lower reactogenicity.

Another technology developed for saRNA delivery is based on liposome-protamine-RNA (LPR) nanoparticles [57]. LPR nanoparticles are prepared using a self-assembly method based on the condensation of saRNA with protamine via electrostatic interactions, followed by cationic liposome encapsulation and surface PEGylation. LPR nanoparticles were further optimized to achieve efficient targeting to dLNs and stimulation of the innate immunity via TLR signaling. This platform was used to deliver a VEEV-based replicon encoding the RBD in mice. Subcutaneous injection of 2 μg of LPR-saRNA in mice induced high and prolonged antigen expression in dLNs, and upregulated multiple innate immune signaling pathways such as IFN-I and STING. Antigen expression levels were significantly higher compared to those of an LPR-mRNA vaccine and lasted for at least two weeks, in contrast to only four days in the case of non-replicating RNA [57].

### 2.3. Beyond Lipids: Other Delivery Strategies for saRNA Vaccines

Despite LNP-mRNA vaccines played a critical role in limiting the pandemic caused by SARS-CoV-2, the difficulties related to their production and distribution emphasize the need for developing improved versions in anticipation of future outbreaks. In addition to lipid-based platforms, other strategies have also been evaluated in the context of saRNA delivery, including the use of naked nucleic acids, vectors based on viral particles (VPs), and even bacteria.

Using naked nucleic acids has the clear advantage of streamlining manufacturing processes and cutting production costs. Although some groups have been able to deliver naked saRNA using different methods, such as electroporation [58,59,60], it would be challenging to achieve efficient vaccination of the population using these approaches due to the high susceptibility of naked RNA to degradation. DNA would be a preferable choice in this scenario; however, more regulatory concerns apply in this case due to potential malignant transformation of transfected cells due to genomic integration. In this regard, the DNA-launched self-amplifying RNA replicon (DREP) is a promising system based on a plasmid that encodes the saRNA under the control of a eukaryotic promoter. Once in the nucleus of the target cell, the saRNA is transcribed from the DNA and exported to the cytoplasm. This platform works in a similar way to saRNA, inducing apoptosis in transfected cells and thus eliminating the unlikely event of chromosomal integration, but with the advantages of DNA stability [61,62]. Using the DREP system, two vaccine candidates encoding either SARS-CoV-2 S protein (DREP-S) or a S trimer ectodomain stabilized in a prefusion conformation were designed and evaluated in mice by homologous and heterologous prime-boost immunization, with a 4-week interval. For immunization, 10 µg of DREP was administered intradermally (ID) followed by local electroporation, using recombinant S protein to boost in the heterologous schedule. All strategies were able to elicit nAb and T cell responses against the S protein. DREP vaccines favored a Th1-biased humoral response either in a homologous or heterologous regimen, in contrast to vaccination with recombinant protein only, which induced predominantly Th2 responses. Interestingly, heterologous immunization using DREP as a prime and recombinant protein as a boost generated the most potent immune responses [63]. Due to the high stability, low costs, ease of production and storage, DREP vaccines could be made readily available on a large scale. This strategy has been proven to be safe and immunogenic in preclinical immunization studies against different pathogens [64,65,66,67] or for cancer therapeutic vaccination [68]. 

In a more recent study based on naked nucleic acid vaccine delivery, Amano and colleagues designed a temperature-controllable saRNA (c-srRNA) from a VEEV vector that was optimized to replicate at 33 °C by an insertion of five amino acids in the sequence of the Rep nsp2 subunit [69]. This property makes it especially suitable for ID administration, where the temperature is between 30–35 °C. The ID route is interesting for vaccination since it is an anatomic site highly rich in antigen presenting cells (APCs), which can uptake saRNA and present the encoded antigens in MHC-I and II to T cells. Interestingly, c-srRNA could be efficiently delivered ID using lactated Ringer’s solution, as demonstrated with a vector encoding the luciferase reporter gene. Expression of luciferase was maintained for nearly a month after one dose of 5 µg of c-srRNA, compared to one week in the case of a nucleotide-modified mRNA. To test the ability of this strategy to induce cellular and humoral immunity against SARS-CoV-2, the sequence of RBD with a signal peptide for secretion was cloned into c-srRNA and mice were immunized with 5 or 25 µg in a prime/boost regimen, 4 weeks apart. The vaccine induced cellular immunity against RBD, characterized by CD8^+^ and Th1 CD4^+^ responses, however, it was unable to elicit a humoral response. Induction of RBD-specific antibodies was only achieved by using a heterologous vaccination scheme, combining the c-srRNA vaccine with recombinant RBD protein. Therefore, although ID c-srRNA vaccine appears to be an exclusively “T cell inducer vaccine”, it can be combined with other strategies to boost humoral responses. Although some efficacy experiments were performed using an artificial model involving tumor cells expressing vaccine antigens, a limitation of the study was the absence of a SARS-CoV-2 challenge in the immunized animals [69].

Bacteria-based vaccines are interesting because they are inexpensive, quickly produced, do not require especial conditions for storage and transport, and could be administered orally, thus avoiding the need for injections. The group of John Hwa Lee has broad experience using *Salmonella* for oral vaccination, and they have demonstrated the potential of this approach using different antigens for immunization. *Salmonella* is an interesting vector due to its ability to infect APCs, such as dendritic cells and macrophages, which can be exploited for vaccination [70]. More recently, this group exploited the use of *Salmonella* for saRNA delivery, using a DNA-launched system based on SFV [61]. With this system the authors achieved efficient expression of four parenteral SARS-CoV-2 antigens (RBD, heptad repeat domain, M protein, and nsp13 epitopes) from a single ORF, in an attempt to generate a potent and broadly neutralizing vaccine. In a first proof-of-concept study, the vaccine candidate was administered IM in mice, demonstrating a good safety profile and eliciting potent immune responses characterized by nAbs, Th1-biased responses, and specific T cells [71]. Next, IM administration of a single dose of 1 × 10^7^ bacterial colony-forming units (CFU) was compared to oral administration of two doses of 1 × 10^8^ CFU with a 2-week interval in mice. After the boost, a robust antibody and cellular response against all antigens and a Th1-biased immunity was induced. The vaccine was also evaluated in Golden Syrian hamsters, which were completely protected against viral infection and lung pathology after challenge with parental SARS-CoV-2. When hamsters were challenged with the Delta VoC, a superior protection was observed in animals immunized orally compared to the IM vaccine. Oral vaccination led to the induction of a significant IgA response in serum and soluble IgA in lung homogenates and intestinal lavage against the different antigens, especially M protein, which could explain its enhanced cross-neutralizing activity [72]. A remarkable protection against Delta variant infection with this oral vaccine was also observed using a simplified mouse model transiently expressing hACE2 in lungs, which was achieved after transduction with an AAV2 encoding this protein [73]. The induction of efficient mucosal immunity is key for preventing the spread of infectious diseases, given that it constitutes the primary point of entry for pathogens [74].

An alternative to cationic lipids to encapsulate RNA is based on the use of polymers. Blakney and colleagues developed a bioreducible, linear, cationic polymer called “pABOL” for saRNA delivery, composed of polyamidoamines. They showed an enhancement of saRNA delivery in vivo and in vitro by increasing the molecular weight of the polymer, and demonstrated the feasibility of using this system in the context of influenza vaccination in mice [75]. The use of pABOL has also been evaluated for SARS-CoV-2 immunization, using a saRNA encoding the S protein. Despite pABOL generating higher levels of transgene expression compared to LNP-formulated saRNA, the latter resulted in higher humoral and cellular immunity in mice, indicating that antigen expression alone is a poor predictor of vaccine efficacy [76].

Finally, the use of viral replicon particles (VRPs) for the delivery of saRNA has also been evaluated in the context of SARS-CoV-2 immunization. In one study, the VEEV replicon encoding the prefusion-stabilized S protein of the Omicron BA.1 variant was packaged into recombinant VRPs, which are devoid of the genes encoding the structural proteins of VEEV and therefore cannot propagate [77]. These VRPs were used to immunize mice and hamsters with three doses every two weeks via the intraperitoneal (IP), IM, and IN routes. Specific anti-RBD IgG and nAbs against the BA.1 variant were induced in all immunized groups with a Th1-biased antibody response, with the IP and IM routes inducing higher antibody titers. Immunized animals were IN challenged with the SARS-CoV-2 Omicron variant to evaluate the efficacy of the vaccine. In mice, which are susceptible to infection by this variant [78], no infectious virus was detected in the lungs after immunization via IP and IM routes. In hamsters, protection against lung pathology was achieved in all immunized animals, with complete protection from viral replication in the lungs in the animals immunized IP [77]. A similar approach using VEEV-derived VRPs expressing an optimized S protein was used to immunize guinea pigs and cats, showing in this last animal model to be able to induce protective immunity preventing SARS-CoV-2 infection and transmission [79]. A summary of the main preclinical studies performed with saRNA COVID-19 vaccines is presented in Table 2.

VRPs have also been used to deliver VEEV-based saRNA encoding monoclonal antibodies with neutralizing activity against SARS-CoV-2 infection, one directed against the RBD of the parenteral Wuhan strain [80] and the second one against the hACE2 receptor [81]. In both cases, the VRPs were administered needle-less via the IN route, and they were able to induce expression of the antibodies in the lung tissue for around five days. In the first study, mice received VRPs and 24 h later were challenged with a mouse-adapted SARS-CoV-2 strain (MP7). The anti-RBD antibody was able to protect animals from weight loss, and decreased viral load and pathology in lungs [80]. The rationale for using an antibody against hACE2 is that, in principle, it would be able to block infection in cells expressing this receptor. Although these strategies may seem irrelevant compared to prophylactic vaccination, they could prove valuable in specific scenarios, such as protecting immunocompromised individuals exposed to infection sources or mitigating viral load and propagation in patients with severe symptoms. Nevertheless, the short-term expression of this system may be insufficient, emphasizing the necessity for optimization of the technology. 

### 2.4. Toxicology Studies

Safety is an important point to take into account when designing saRNA vaccines, since immunostimulatory properties can be modulated by its different components, i.e., the delivery vehicle, RNA self-amplification itself, and antigen selection. Biodistribution and toxicology studies for some SARS-CoV-2 saRNA vaccines have been performed in different models. Sprague-Dawley rats, a species that is routinely used for toxicology studies of vaccines, is of particular interest in the context of saRNA-based vaccines because, like humans, they express TLR7, an important receptor for the recognition of exogenous RNA [82]. In the previously mentioned work by Maruggi et al., the authors evaluated toxicity and biodistribution of SARS-CoV-2 saRNA-LNPs vaccine in rats. A repeated-dose toxicity study was conducted in which animals received three IM doses at two-week intervals. The vaccine was well-tolerated, although transient changes were observed, such as an increase in rectal temperatures, local erythema after immunization, edema after the second and/or third dose in some animals, and transient changes of hematology parameters. However, these effects were resolved in 48–72 h, with no changes in body weight. Regarding biodistribution, the tissues with higher saRNA levels were muscle, lymph nodes, and spleen, with detectable levels up to day 60 after a single dose of 6 µg. A transient dissemination to distal sites was also observed, including heart, liver, kidney, lung, and blood, although levels were lower and decreased more rapidly [38].

The additional effects of the RNA self-amplification and the composition of the delivery vehicle were evaluated in another study, in which LNPs were compared to a cationic nano-emulsion (CNE) [83]. This CNE, composed of the cationic lipid DOTAP, has previously been used to deliver saRNA-based vaccines in different animal models, and demonstrated to be immunogenic and well-tolerated [84]. Rats received two doses of 12 µg of a saRNA encoding S protein with a two-week interval. Overall, no safety concerns were observed for any of the vaccines besides a mild transient temperature elevation and localized inflammation. LNP-based vaccines appeared to activate innate immune responses earlier than the CNE vaccine, with higher TNF-α concentrations in serum at four hours and upregulation of more innate immunity-related genes, primarily associated to type I/II IFN signaling pathways. This effect was transient and mediated by the vehicle rather than by RNA. These subtle differences in innate immune stimulation should be taken into consideration since they could have an impact on safety and efficacy when evaluated in the clinic. 

**Table 2 vaccines-12-00318-t002:** saRNA vaccines tested in representative preclinical studies.

Vector	Vaccine Formulation	Encoded Antigen	Vaccination Schedule	Animal Model	Main Results	Ref.
VEEV	LNP	Prefusion stabilized S protein	IM, 0.015–1.5 μg, prime/boost (3-week gap)	Mouse	nAbs, T cell response (Th1)	[38]
			IM, 0.03–3 μg, prime/boost (3-week gap)	Hamster	Partial protection against infection	[38]
			IM, 0.01–10 μg, prime/boost (4-week gap)	Mouse	nAbs, T cell response (Th1)	[40]
			IM, 5 μg, prime/boost (4-week gap)	Hamster	Protection from severe disease	[41]
		Unmodified protein	IM, 0.2–10 μg, only prime	Mouse	nAbs, T cell response (Th1)	[42]
				hACE2 Mouse	Protection against infection	
		RBD-TM (Wuhan- Hu-1)	IM, 1 μg, prime/boost (4-week gap)	Mouse	Cross-reactive Abs, specific CD4^+^ and CD8^+^ T-cell responses	[85]
			IM, 50 μg, prime/boost (4-week gap)	Cynomolgus monkey	Protection against infection	
		RBD-TM (gamma)	IM, 10 μg, prime/boost (4-week gap)	Hamster	Cross-nAbs, antigen-specific B and T cells, prevent weight loss	
	NLC	Prefusion stabilized protein	IM, 1–30 μg, only prime or prime/boost (3-week gap)	Mouse	Cross-nAbs, Th1 response	[46]
	LION	Unmodified protein	IM, 0.1–10 μg, prime/boost (4-week gap)		nAbs, T cell responses (Th1)	[47]
			IM, 250 μg only prime or 50 μg prime/boost (4-week gap)	Pigtail macaque	nAbs, modest T cell responses	
			IM, 5 to 50 μg, prime/boost (4 to 20-week gap)		nAbs, partial protection from infection, protection from disease	[48]
			IM, 5 or 25 μg, only prime	SIV-infected pigtail macaque	nAbs, modest T cell responses	[49]
		S protein from different VoCs	IM, 1 μg, prime/boost (4-week gap)	Mouse	Cross-nAbs	[53]
			IM, 20 μg, prime/boost (4-week gap)	Hamster	Cross-nAbs, partial protection from infection and disease	
	LPR	RBD	SC, 2 μg, prime/boost (4- week gap)	Mouse	nAbs, extended antigen expression in dLNs	[57]
	LNP + Ad	Full-length S protein	IM, 10^8^ i.u. Ad prime, and 1μg RNA boost (4-week gap)	Mouse	nAbs, cytotoxic T cells and Th1	[86]
		Codon-optimized S protein	SC, 10 μg, prime/boost (8-week gap)	Mouse	nAbs, specific T-cell responses	[87]
			IM bilateral, 3–30 μg, prime/boost (4-week gap)	Rhesus macaque		
	LNP + OX40 agonist	Prefusion stabili- zed trimeric S protein	IM, 1 μg, prime/boost (4-week gap)	Mouse	Specific CD4^+^ and CD8^+^ T-cell responses	[88]
	VRP	Prefusion stabilized S protein (Omicron)	IP, IM or IN, 1 × 10^6^ VRPs, prime/two boosts (2-week gap)		nAbs, Th1 (IP route), protection against infection	[77]
				Hamster	nAbs, Th1, protection against disease	
		Optimized S protein	IM, 10^7^ VRPs, prime/two boosts (3-week gap)	Guinea pig	nAbs	[79]
			SC, 5 × 10^7^ VRPs, prime/boost (3-week gap)	Cat	nAbs, protection against infection and transmission	
DNA/SFV replicon	Naked, Ep	Unmodified or prefusion stabilized S protein	ID, 10 μg, prime/boost (4-week gap)	Mouse	nAbs, T cell response (Th1)	[63]
	Salmonella	RBD, HR, M, nsp13	IM, 1 × 10⁷ CFU, only prime, or oral, 1 × 10⁸ CFU, prime/boost (2-week gap)		nAbs, specific CD4^+^ and CD8^+^ T-cell responses	[71]
			IM, 2 × 10⁷ CFU, only prime, or oral, 2 × 10⁸ CFU, prime/boost (2-week gap)	Hamster	Cross-nAbs, IgA, protection against infection and disease	

Abs, antibodies; Ad, adenovirus; CFU, colony forming units; dLNs, draining lymph nodes; Ep, electroporation; HR, heptad repeat domain; ID, intradermal; IM, intramuscular; IN, intranasal; IP, intraperitoneal; i.u., infectious units; LION, Lipid InOrganic Nanoparticles; LNP, lipid nanoparticle; LPR, liposome-protamine-RNA; M, membrane; nAbs, neutralizing antibodies; NLC, nanostructured lipid carrier; nsp, non-structural protein; RBD, receptor binding domain; S, spike protein; SC, subcutaneous; SFV, Semliki Forest virus; SIV, simian immunodeficiency virus; TM, transmembrane; VEEV, Venezuelan Equine Encephalitis virus; VRPs, viral replicon particles.

## 3. Strategies to Increase Efficacy of saRNA Vaccines

Despite some groups having shown that saRNA vaccine candidates can induce immunity with high magnitude and breadth, an important concern is the emergence of new VoCs that could evade the immunity induced by the current vaccines. Therefore, efforts are being made to generate vaccines inducing a broader neutralizing response against multiple VoCs. 

### 3.1. Multi-Antigenic Vaccines 

One strategy is to design multi-antigenic vaccines that elicit both humoral and cellular responses, since cell-mediated immunity could have an important role in viral clearance. Cellular immune responses against conserved antigens, such as N protein, could increase the protection against other VoCs, particularly in the context of waning antibody responses. ZIP1642 is an LNP-formulated dual antigen vaccine encoding the S-RBD and N antigens of the Wuhan strain in two independent saRNAs [89]. In mice immunized with 1 µg of total saRNA (1:1 ratio of each RNA) in a prime-boost schedule with a 21-day interval, this vaccine was able to induce nAbs against four VoCs, expand S- and N-specific CD4^+^ and CD8^+^ T cells, and induce a Th1 cytokine profile. In addition, hamsters vaccinated with ZIP1642 following a prime-boost schedule with doses of 1 or 5 µg generated a significant nAb response. Four days after IN challenge with a Wuhan-like variant, these animals showed significantly lower viral loads in lungs compared to unvaccinated controls, and the highest dose was able to protect animals from virus-induced lung pathology. When hamsters were challenged with the Beta B.1.351 variant, this vaccine was also able to decrease the viral load in lungs and protect them from weight loss, although lung pathology was not improved compared to controls. In this case, no significant nAb responses against the Beta B.1.351 variant were elicited by the vaccine, suggesting that its action is via enhanced cellular immunity or non-nAb functions [89]. 

### 3.2. Optimization of Antigen Selection

It has been reported that after SARS-CoV-2 infection, more than 90% of nAbs are directed towards the RBD of S protein [90]. Following this standpoint, Komori et al. designed saRNA vaccines expressing soluble RBD or a version of this protein anchored to the membrane (RBD-TM) [85]. In preliminary experiments in mice, significantly higher antibody titers as well as cellular responses against several VoCs were induced by RBD-TM compared to its secreted counterpart. The saRNA vaccine expressing RBD-TM was subsequently tested in hamsters and NHPs, leading to high immunogenicity and protection against virus challenge. To augment the immunogenicity and broaden the protection conferred by this vaccine, a new saRNA expressing RBD from the SARS-CoV-2 Gamma variant fused to TM was generated and tested in hamsters. Vaccinated animals were protected against weight loss after challenge with the Gamma variant, outperforming the original saRNA RBD-TM vaccine based on the Wuhan-Hu-1 sequence [85]. Modifying the RBD sequence would be sufficient to update the saRNA-RBD-TM vaccine in response to emerging VoCs, potentially simplifying the development process of new vaccines.

Although most COVID-19 RNA vaccines are based on the expression of either the full-length S protein or its RBD domain, several groups have proposed, as an alternative, the use of a few S protein epitopes to generate new candidate vaccines. These epitopes can be selected based on their capacity to trigger B or T cell responses and can be adapted to the most prevalent HLA alleles in the population. An additional advantage of epitope-based vaccines is that the polypeptide to be expressed could be considerably smaller compared to the complete S protein, something that might increase its expression levels, potentially allowing the use of lower doses of saRNA. Two different studies have used this approach to design saRNA COVID-19 vaccines [91,92]. In both cases the authors used immunoinformatics to predict immunodominant regions of the S protein, selecting either three hub regions of about 100 residues, each covering the largest number of overlapping B- and T-cell epitopes [91], or using twelve different B- and T-cell epitopes fused by small linkers, resulting in a vaccine polypeptide of 196 residues [92]. Although the authors performed in silico predictions to evaluate safety, stability, and immunogenicity of these vaccines, they did not test them in vivo, something that limits their possible clinical use.

### 3.3. Optimization of saRNA Molecules

A recent finding that could constitute a major breakthrough for the development of saRNA vaccines is the discovery that this type of RNA can also be synthetized using modified nucleotides. Until now, it was believed that this was not feasible, since many groups had failed to modify saRNA with pseudouridine, the most common modification used in conventional mRNA vaccines. However, two recent reports showed that saRNA can be modified with other nucleotides, such as 5-hydroxymethylcytidine, 5-methylcytidine (5-mC), and 5-methyluridine [93,94]. These modifications led to a considerable reduction in IFN-I induction both in vitro and in vivo, resulting in a higher and more robust expression of transgenes compared to unmodified saRNA. The reduction of IFN-I production mainly affected plasmacytoid dendritic cells (pDCs), which seemed to be the main source of this cytokine in response to saRNA [93]. In fact, one of these studies showed that a fully 5-mC modified saRNA vaccine led to significant protection in mice against a lethal challenge with a mouse-adapted SARS-CoV-2 strain using a 100-fold lower dose than a modified mRNA vaccine [94]. Interestingly, and despite being so recently developed, a 5-mC-modified saRNA expressing a membrane-anchored RBD domain of SARS-CoV-2 spike protein has already been tested in a phase I clinical trial in Japan (jRCT2051230005). This LNP-encapsulated vector, termed VLPCOV-02, was able to boost specific IgG responses with a dose as low as 1 µg [95]. Interestingly, and although not compared side by side with its non-modified counterpart, VLPCOV-02 appeared to be less reactogenic, as suggested by lower fever rates in participants receiving the same dose of these vaccines.

### 3.4. Heterologous Vaccination Regimens

Due to the rapid development of vaccines against SARS-CoV-2 and their limited availability during the pandemic outbreak, it was likely to encounter scenarios in which individuals had received prime and boosts from different approved vaccines. However, heterologous vaccination regimens may present different safety or efficacy profiles. In fact, clinical studies evaluating ChAd-mRNA heterologous vaccination have shown that this regimen can induce higher frequencies of S-specific T cells and higher nAb titers against different VoCs compared to homologous vaccination [96].

The added benefit of mixed modality vaccinations has also been evaluated for saRNA-based vaccines, either using two different saRNAs or combining saRNA with a different vaccine platform. Heterologous vaccination using adenovirus ChAdOx1 and a saRNA-LNP vaccine in mice induced higher antibody titers and a superior cellular immune response, compared to homologous vaccination regimens [86]. Similarly, a ChAd prime followed by a boost using saRNA-LNP encoding prefusion stabilized S-protein was tested in rhesus macaques. Robust T-cell and nAb responses were achieved with this regimen, which protected animals from SARS-CoV-2 replication post-infection, although homologous vaccination using a saRNA vaccine elicited similar responses at different doses [87]. This heterologous regimen is currently being evaluated in a clinical trial in individuals that have been primed with adenoviral vaccines (NCT04776317). 

Interestingly, another study showed an enhancement of immune responses using the saRNA and adenoviral vaccines in the reverse order, i.e., priming with saRNA and boosting with adenovirus [97]. The authors observed enhanced specific CD4^+^ and CD8^+^ T-cell responses in mice following this heterologous regimen, suggesting that the priming effect of the saRNA may promote efficient CD4^+^ T cell activation and create conditions for a more robust CD8^+^ T-cell response upon adenoviral vaccine boost.

Given the fact that a great proportion of the world population already presents antibodies against SARS-CoV-2, acquired through vaccination or infection, a heterologous boost could be the most effective approach to enhance responses in the presence of a pre-existing immunity to SARS-CoV-2. This is particularly interesting in the case of individuals that have received adenovirus-based vaccines since immune responses against the vector itself could diminish efficacy upon re-dosing.

### 3.5. Combination with Immunostimulatory Molecules

The immune responses induced by vaccination could be enhanced by a combination of vaccines with molecules that can activate immunostimulatory receptors on immune cells, such as OX40, CD137, or CD40, or with molecules that can block immune checkpoints such as PD-1, PD-L1, and CTLA-4, among others. In the case of saRNA vaccines, this type of combinatory approach has been tested by concomitant OX40 activation. OX40 agonist antibodies or its ligand (OX40L) can induce higher levels of T cell cytokines, aid in the clearance of viruses, and enhance antitumor T cell responses. Following antigen recognition, OX40 undergoes upregulation on CD4^+^ T cells and, to a lesser extent, on CD8^+^ T cells. After binding to OX40L, naturally found on activated APCs, T cells display enhanced survival, expansion, and effector functions [88]. Considering this, and with the purpose of extending the duration of the vaccine-induced immune response, Duhen and colleagues employed an OX40 agonist alongside a saRNA encoding the S protein, formulated in LNPs, to immunize mice. This study found that the OX40 agonist increased specific CD4^+^ T cell responses, for both protein and saRNA vaccines. Interestingly, the saRNA-based vaccine generated a stronger S-specific CD8^+^ T cell response compared to protein-based vaccination. In addition, these responses were further amplified by the co-administration of the OX40 agonist [88].

## 4. saRNA-Based COVID-19 Vaccines in Clinical Trials

Despite the fact that the first mRNA vaccines were approved only three years ago, many saRNA vaccine prototypes have already reached clinical trials. Preliminary studies in humans showed that saRNA is generally safe and can be used at lower doses compared to mRNA vaccines, although this observation has not been consistent across all trials. One possible explanation for this variability is that, until recently, saRNA vaccines could not be modified with nucleotide analogues, which made them more susceptible to IFN-I responses. As discussed in Section 3.3, the recent discovery that saRNA can be synthetized using certain modified nucleotides, such as 5-mC, could open new possibilities to develop these vaccines.

### 4.1. Vaccines Based on S Protein

Among the first saRNA vaccine trials is a phase I performed in the UK with the so-called COVAC1 vaccine, based on LNPs containing a VEEV-based saRNA that expresses S protein (LNP-nCoVsaRNA). The study assessed vaccine doses from 0.1 μg to 10 μg, given IM, with a four-week interval between prime and boost. COVAC1 exhibited a safety profile similar to other COVID-19 mRNA vaccines, with common systemic and local reactions, particularly among younger adults. Disappointingly, seroconversion rates ranged from 39% to 61% for doses between 1 and 10 μg [55]. 

A phase II trial of COVAC1 was conducted in the UK to expand the safety and immunogenicity study of this vaccine [98]. In this case, a prolonged interval of 14 weeks was maintained between prime and boost, with a 1 μg dose followed by a 10 μg dose. Principio del formulario

Tolerability was dose-dependent, with a higher frequency and severity of adverse reactions observed after the higher dose. A higher rate of seroconversion was observed, reaching 80% regardless of age. After priming with 1 μg, participants with a history of COVID exhibited an antibody response comparable to a third booster with other COVID-19 vaccines.

Since the seroconversion rate in both studies was lower compared to other SARS-CoV-2 vaccines, a second-generation vaccine (LNP-nCoV saRNA-02) was developed to address these limitations. This new vaccine, with an undisclosed redesigned saRNA backbone to dampen IFN responses, will be used in a phase I trial in Uganda (NCT04934111) [99]. 

A different saRNA vaccine candidate tested in a phase I/II trial is ARCT-021 (LUNAR-COV-19) [54]. The ARCT-021 vaccine contains a VEEV–derived saRNA expressing a codon-optimized S protein formulated in patented LUNAR LNPs [42]. In the phase I study, ascending levels of one-dose ARCT-021 (1, 5, 7.5, and 10 μg) were administered. In contrast to COVAC1 trial, the seroconversion rate was between 80 and 100% in all cohorts. Phase II tested ARCT-021 in a prime-boost regimen using doses of 3 or 5 μg, given 28 days apart. These trials showed that ARCT-021 is immunogenic and has a favorable safety profile. Additional findings included the generation of T cell responses against the S protein, characterized by a Th1-biased phenotype, and the fact that a second dose did not result in a significant increase in nAb titers against SARS-CoV-2 [54]. 

Although innate immunity plays a key role in shaping the adaptive immunity in response to vaccination [100,101], it is known that overactivation of innate immune responses, particularly IFN-I, may prematurely hinder the translation of saRNA and diminish the immunogenicity of this vaccine [102,103]. Since molecular signatures induced early after vaccination can correlate and predict the later adaptive immune responses [100], the immune transcriptional profile in blood was studied during the phase I/II clinical trial for the ARCT-021 vaccine, and compared to data from participants vaccinated with BNT162b2 or other more studied forms of vaccines [104]. Maximal transcriptional changes in innate immune genes were observed at day two postvaccination, and the magnitude of these changes correlated positively with anti-S IgG titers at day 29. Transcripts involved in T responses became apparent only at day eight, with genes related to T cell maturation correlating positively with S-reactive T-cell responses at day 15. Similar transcriptional signatures at day one after the first dose were observed for ARCT-021 and other vaccines, such as viral vectors, adjuvanted vaccines, and the mRNA vaccine BNT162b2. However, no significant increase in innate immune activation was observed after the second dose of ARCT-021, in contrast to BNT162b2. This could be due to overactivation of innate immune response by saRNA, leading to reduction of the boost effectiveness. These observations may help fine-tune saRNA vaccines to avoid prematurely triggering host immune responses, allowing a prolonged antigen expression, and increasing immunogenicity [105].

An improved version of ARCT-021 targeting VoCs (ARCT-154) has been recently compared to the mRNA vaccine BNT162b2 in a phase III trial [106]. Both ARCT-154 and BNT162b2 were given as a fourth booster to volunteers previously vaccinated with mRNA vaccines (BNT162b2 or mRNA-1273). On day 28 after vaccination, ARCT-154 induced immune responses that were comparable to BNT162b2 against the Wuhan-Hu-1 strain but superior against the Omicron BA.4/5 variant. Therefore, ARCT-154 could potentially offer enhanced protection against VoCs and an extended duration of immunity. Based on these results, the ARCT-154 vaccine received authorization in Japan on 27 November 2023 [21].

In general, saRNA vaccines based on LNPs have shown a certain degree of reactogenicity associated to the dose, with a higher proportion of local reactions and adverse effects in those individuals receiving 10 μg, but with a very good tolerability when using doses of 1 µg or less [55]. An inverse association of reactogenicity and age has also been described, with adverse reactions being less frequent at older ages [98]. It is possible that the use of delivery vehicles different from LNPs, such as LION, could reduce the reactogenicity of these vaccines [56]. In fact, a COVID-19 saRNA vaccine based on LION has recently received emergency licensure in India based on a phase II/III clinical trial (CTRI/2021/09/036379), but results regarding the efficacy and safety of this vaccine have not been published yet [107].

### 4.2. Vaccines Based on the RBD

A saRNA vaccine expressing membrane-anchored RBD, as described in Section 3.2, has also been evaluated in a phase I trial. This vaccine, named VLPCOV-01, was evaluated in healthy participants that had previously received two doses of BNT162b2. The study revealed robust immune responses in both non-elderly and elderly healthy adult participants, even with a dose of 0.3 µg. Elevated IgG levels were sustained for 13 weeks in saRNA groups, with T cell responses comparable to those elicited by BNT162b2, particularly in the CD8^+^ compartment. Furthermore, participants exhibited nAb responses against several VoCs [108]. These findings strongly suggest that a low dose of VLPCOV-01 vaccine maintains a favorable safety profile while inducing immune responses comparable to those triggered by the BNT162b2 mRNA vaccine.

### 4.3. Multi-Antigenic Vaccines

GRT-R910 is a novel saRNA vaccine formulated into LNPs, encoding prefusion S protein as well as conserved non-S T cell predicted epitopes derived from N, M, and ORF3a genes. A phase I clinical trial in the UK evaluated the boosting effect of the GRT-R910 vaccine among healthy adults aged over 60, who had received one or two doses of the adenovirus AZD1222 vaccine. GRT-R910 was well-tolerated, even at the highest dose (30 µg) and induced an increase in both binding and nAbs to Wuhan Hu-1 S protein and other VoCs such as Omicron BA1 and BA4/5, with nAb titers that were sustained for six months. In addition, GRT-R910 boosted S-specific T cell responses primed by AZD1222, and generated de novo T cell responses against non-S epitopes. Despite a limited sample size, preliminary data suggest that GRT-R910 is able to induce a long-term immune response in an older population, potentially paving the way for a dose-sparing vaccine platform [109].

### 4.4. Heterologous Vaccination

As outlined in Section 3.4, incorporating heterologous vaccination regimens may result in favorable outcomes. In a recent clinical trial, a heterologous vaccination strategy combining saRNA with mRNA was compared to homologous mRNA vaccination. Participants received either two doses of a saRNA vaccine plus two doses of an approved COVID-19 mRNA vaccine (BNT162b2), or only two doses of BNT162b2. Surprisingly, there were no significant differences in humoral or cellular responses between groups. However, upon categorizing participants based on prior COVID-19 status, the researchers noted that two weeks after the second dose of the mRNA vaccine, nAb titers were notably higher among saRNA recipients with a history of COVID-19 compared to those who received homologous mRNA vaccination or COVID-19-naive saRNA recipients [110]. These findings suggest that increased antigen exposure, whether from natural infection or vaccination, may confer immunological advantages that are particularly evident in individuals undergoing heterologous vaccination. A summary of clinical trials for COVID-19 vaccination using saRNA is presented in Table 3.

## 5. Conclusions and Future Directions

In 2020, the field of vaccination experienced a revolution with the development and authorization of the first mRNA vaccines that aimed to protect from SARS-CoV-2 infection. To date, these vaccines have already been used to immunize hundreds of millions of people against COVID-19 with an excellent efficacy and safety profile. This remarkable achievement was feasible thanks to the possibility to synthetize mRNA with modified nucleotides and to the development of optimal lipid formulations that allow efficient transduction of mRNA in vivo without toxicity. In parallel to conventional mRNA vaccines, many laboratories had previously studied the possibility of using saRNA for vaccination. In fact, as early as 1998, the group of Peter Liljeström at the Karolinska Institute (Sweden) showed that mice could be vaccinated with naked saRNA derived from SFV [62]. One advantage of saRNA over conventional mRNA lies in its self-replicating nature, which can decrease the required RNA quantity for vaccination, since even if only one or a few saRNA molecules enter a cell, they could undergo rapid amplification by several orders of magnitude in a short period of time. In contrast, for conventional mRNAs, several molecules per cell are probably needed to achieve a reasonable antigen expression. On the other hand, as shown by Katalin Karikó and Drew Weissman, mRNA needs to be modified to avoid deleterious IFN responses that can lead to its early degradation [10]. Until recently, it was believed that nucleotide modifications were not necessary for saRNA due to its rapid amplification and to the induction of translational shut-off of endogenous proteins in transfected cells, which also affect the IFN pathways. But this was an assumption that had not been tested due to the fact that saRNA does not tolerate the incorporation of the most commonly modified nucleotides used in conventional mRNAs, such as pseudouridine. However, this scenario has recently changed with the discovery that saRNA can be modified with certain nucleotides, like 5-mC, and that these modifications improve the performance of these vectors [93,94]. Remarkably, one of these modified saRNAs has already been tested in a phase I clinical trial in Japan with promising results [95]. 

The incorporation of saRNA in vaccine development has greatly benefited from the use of optimized LNP formulations for in vivo delivery, although some alternatives to LNPs may offer additional advantages in the context of saRNA delivery as described in Section 2.2. Nevertheless, the LNP platform has allowed the comparison of saRNA COVID-19 vaccines with conventional mRNAs side by side in preclinical studies performed in animal models susceptible to SARS-CoV-2 infection and pathology. Most of these studies, which have been covered in this review, showed that saRNA could be as efficient as mRNA for immunization but with the additional advantage that it enables the use of lower doses, potentially reducing the cost of vaccines and lowering the occurrence of minor adverse events. Despite these promising results in preclinical studies, most of the data obtained from the few clinical trials performed to date have not shown a clear advantage of saRNA over mRNA, although it is true that good seroconversion rates have been observed in some of these trials with very low doses of saRNA [55]. One problem faced by clinical trials evaluating saRNA vaccines is the fact that participants had been previously vaccinated with authorized vaccines, which means that only a boosting effect can be measured. Despite these difficulties, some of the formulations showed saRNA to be superior for boosting compared to mRNA, leading to the recent approval of saRNA vaccines in India and Japan [21,111]. While more approvals of saRNA-based vaccines will likely take place in the near future, it is possible that some improvements will be needed to enhance their competitiveness against conventional mRNA vaccines.

Some studies suggest that one problem that saRNA vaccines face when used for priming is that overactivation of innate immune responses induced by the self-replicating RNA can hamper the effect of a boost [104]. In this sense, it has been proposed that modifications in the RNA that allow longer antigen expression, for example by introducing mutations in the replicase gene or co-expression of antiapoptotic factors from the same vector, could improve the performance of these vaccines [112,113]. In addition, optimizing antigen selection is likely to be beneficial, favoring shorter, more easily expressed sequences like the RBD over the full-length S protein. The inclusion of fewer epitopes in the vaccine design reduces the concern of antibody-dependent enhancement (ADE) induction, a phenomenon that is primarily mediated by non-neutralizing antibodies [114,115]. Furthermore, this approach can also simplify the combination of several RBD domains from different VoCs into a single multivalent vaccine, by using multiple subgenomic promoters within the same saRNA molecule. 

The potent adjuvant effects of RNA replication and their low-dose requirement are likely to prompt the approval of additional saRNA vaccines not only for COVID-19 but also for other infectious diseases and even cancer, with the potential to become a second revolution in the field of mRNA vaccines.

## Figures and Tables

**Figure 1 vaccines-12-00318-f001:**
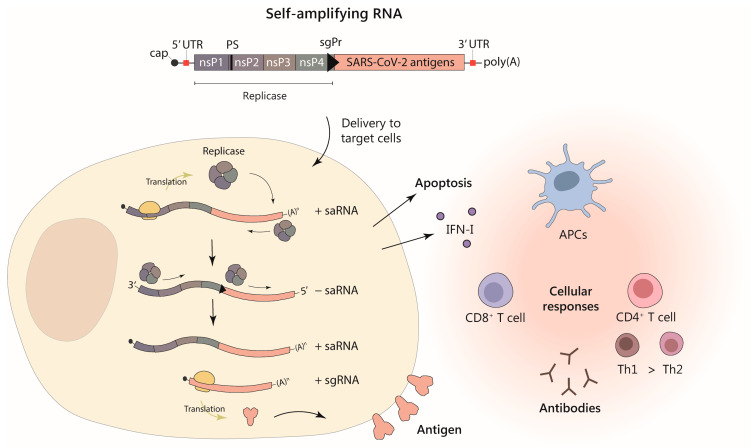
Alphavirus-based self-amplifying RNA vector expressing SARS-CoV-2 antigens for vaccination. The upper diagram represents the saRNA, which is a single strand positive-sense RNA containing a 5′ methylguanylate cap structure (cap) and a 3′ polyadenylate sequence (polyA). The saRNA vector contains two open reading frames (ORFs): the first one codes for the viral replicase, containing four subunits, or non-structural proteins (nsP); the second ORF codes for the SARS-CoV-2 antigen(s). saRNA also includes sequences necessary for replication (5′and 3′ untranslated regions, UTR), a packaging signal (PS), and a subgenomic promoter (sgPr) between both ORFs. The lower left part of the figure represents the replication and expression of saRNA in transduced/transfected cells. Once the saRNA reaches the cytoplasm of a target cell, the replicase is translated from the first ORF. The replicase synthesizes the complementary negative-sense strand of the saRNA (− saRNA), which is later used by the replicase as template to generate more saRNA (+ saRNA) in a self-amplification process. In addition, the replicase can recognize the sgPr in the negative saRNA strand, synthesizing a smaller subgenomic RNA (sgRNA) of positive polarity containing the second ORF, which will be translated to produce the desired antigen (represented on the surface of the cell). With this system, high levels of expression of the vaccine antigen(s) are achieved, as well as induction of type I interferon (IFN-I) responses and apoptosis due to saRNA replication. In addition, this approach can induce specific humoral (antibodies) and cellular immune responses (CD4^+^ and CD8^+^ T cells) against SARS-CoV-2, mediated by the presentation of antigens both on the surface of transduced/transfected cells and by antigen presenting cells (APCs), which can uptake antigens from apoptotic cells (right part of lower figure).

**Figure 2 vaccines-12-00318-f002:**
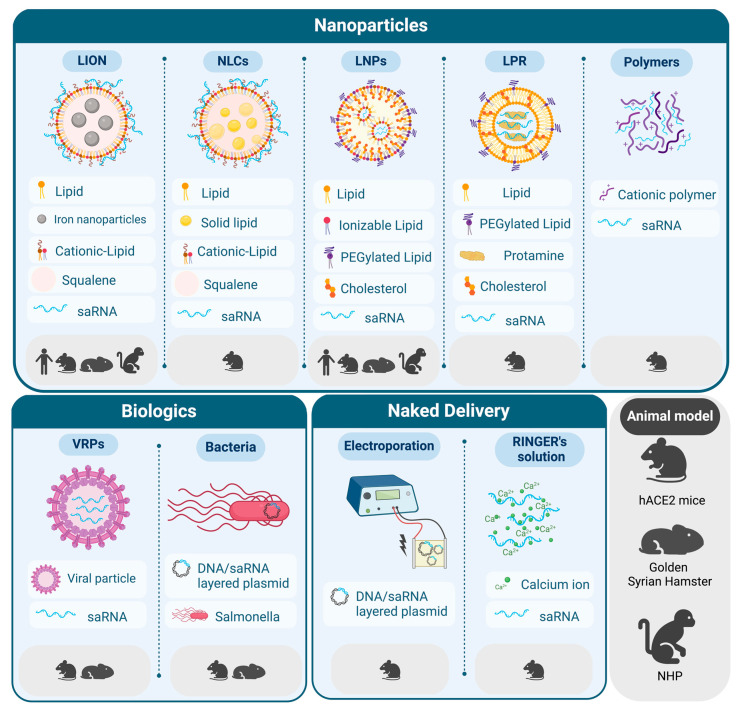
Modalities to deliver saRNA vaccines. Delivery methods include nanoparticles (upper panel), biological vectors (Biologics, lower left panel), and delivery of naked RNA (Naked delivery, lower central panel). The components of each delivery vehicle are indicated in each modality, as well as the main animal models in which they have been tested (animal icons). Preclinical models include Golden Syrian Hamsters, non-human primates (NHP), and transgenic mice either expressing hACE2 or transduced with adeno-associated virus or adenoviral vectors encoding hACE2 (hACE2 mice) (lower right panel). A human icon indicates that the vaccine has also been tested in clinical trials. LNPs (lipid nanoparticles), LION (Lipid InOrganic Nanoparticle), LPR (liposome-protamine-RNA nanoparticle), NLCs (nanostructured lipid carrier), VRPs (viral replicon particles). Created with BioRender.com.

**Table 3 vaccines-12-00318-t003:** Clinical trials evaluating saRNA-based COVID-19 vaccines.

Vaccine Name	Clinical Trial Type	Vaccine Formulation	Encoded Antigen	Administration Schedule	Main Results	Ref.
LNP-nCoV saRNA (COVAC1)	Phase I	LNP	Prefusion stabilized S protein	IM, 0.1–10 μg, prime and boost, 4-week gap	39–61% seroconversion rates. Reactogenicity with 10 μg dose	[55]
	Phase II			IM, 1 μg (prime) & 10 μg (boost), 14-week gap	80% seroconversion rates	[98]
	Phase I			IM, 1 μg (prime) & 10 μg (boost), 14-week gap followed by two doses of licenced vaccine (BNT162b2 or AZD1222)	Higher nAb titers among saRNA recipients with a history of COVID-19 compared to licenced vaccines	[110]
LNP-nCoV saRNA-02				IM, 5 μg, prime and boost, 4-week gap	Ongoing trial	[99]
ARCT-021 (LUNAR-COV-19)		LUNAR LNP	Codon-optimized S protein	IM, 1–10 μg, one dose	80–100% seroconversion rate. Robust Th1 type responses	[54]
	Phase II			IM, 3 and 5 μg, prime and boost, 4-week gap	After second dose, no rise in nAbs	[54]
ARCT-154	Phase III		S protein (D614G variant)	IM, fourth-dose boost after licenced vaccine (BNT162b2 or mRNA-1273)	Immune responses comparable or superior to BNT162b2 against Omicron BA.4/5 variant. Authorized in Japan in 2023	[106]
GRT-R910	Phase I	LNP	Prefusion S (Wuhan Hu-1) and N, M & ORF3 T cell epitopes	IM, 10–30 μg, one dose boost after licenced vaccine (AZD1222, one or two doses)	Binding Abs and nAbs against the original strain and VoCs. Boosted AZD1222-induced T cell responses	[109]
VLPCOV-01			RBD-TM	IM, 0.3 or 1 μg, one dose boost after licenced vaccine (BNT162b2)	Sustained immune responses comparable to BNT162b2. CD4^+^ CD8^+^ T-cell responses. nAbs against VoCs	[108]
VLPCOV-02		LNP (5-mC modified saRNA)	RBD-TM (gamma VoC)	IM, 1–15 μg, one dose boost after licenced vaccine (BNT162b2)	Immune responses comparable to BNT162b2. CD4^+^ and CD8^+^ T-cell responses. and nAbs against VoCs	[95]

AZD1222, Oxford–AstraZeneca Ad vaccine; BNT162b2, Pfizer-BioNtech mRNA vaccine; mRNA-1273, Moderna mRNA vaccine Spikevax.

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
