# Peer review of "Self-Amplifying RNA: A Second Revolution of mRNA Vaccines against COVID-19"

_vaccines, 2024, doi:10.3390/vaccines12030318_

Round 1

Reviewer 1 Report

Comments and Suggestions for Authors

Minor comments:

 Line 20 – in this reviewer’s opinion, “efficient” should be replaced with effective or protective. Eliminate the word “very”.

Line 200, figure 2 – depicted is misspelled.

Line 540 – composed of (not by) the cationic lipid DOTAP

Line 558, Table 1 – suggest adding the dose for each vaccine.

Line 558, Table 1 – ensure that all rows have complete information. Information is missing for the following:

               LPN – reference 29 and RBD-TM (gamma)

               LION – references 37 and 38

               LNP+aOX40 – add antigen (S protein)

               VP – reference 70 is for mouse and hamster

Line 817, Table 2 – ensure that each administration schedule is complete to include the route, dose, schedule.

               Reference 88 – add route

               Reference 96 – add dose amount

               Reference 98 – add dose amount

               Reference 85 – ug is duplicated

Tables 1 and 2 – add the variant for the antigen

Author Response

Dear Editor and Reviewers,

The review manuscript entitled “Self-amplifying RNA: a second revolution of mRNA vaccines against Covid-19” has been revised according to the Reviewer´s suggestions. Please find below a point-by-point reply to each of your comments. All the new data and changes suggested by you have also been highlighted in red in the manuscript.

Reviewer 1:

Line 20 – in this reviewer’s opinion, “efficient” should be replaced with effective or protective. Eliminate the word “very”.

We agree with this reviewer, the word has been changed.

Line 200, figure 2 – depicted is misspelled.

Corrected. Thanks for noticing it

Line 540 – composed of (not by) the cationic lipid DOTAP

Corrected.

Line 558, Table 1 – suggest adding the dose for each vaccine.

The dose of each vaccine has been added in Tables 2 and 3 (previously tables 1 and 2)

Line 558, Table 1 – ensure that all rows have complete information. Information is missing for the following:

               LPN – reference 29 and RBD-TM (gamma)

               LION – references 37 and 38

               LNP+aOX40 – add antigen (S protein)

               VP – reference 70 is for mouse and hamster

We have completed the requested information (please notice that when the same information is repeated in several contigous rows of the same table it only appears in the first row).

Line 817, Table 2 – ensure that each administration schedule is complete to include the route, dose, schedule.

               Reference 88 – add route

               Reference 96 – add dose amount

               Reference 98 – add dose amount

               Reference 85 – ug is duplicated

Tables 1 and 2 – add the variant for the antigen

We have completed the requested information (we have added dose and routes for all examples, as well as antigen variant when needed).

Reviewer 2 Report

Comments and Suggestions for Authors

Manuscript title: Self-amplifying RNA: a second revolution of mRNA vaccines against Covid-19

The authors have started their introduction with the COVID-19 pandemic and continued focus on the self-amplifying RNA, including different delivery systems, other than LNPs, viral vector delivery systems, toxicological aspects, S-based vaccines, heterologous prime/boost strategies, and finally successful clinical trials. The conclusion is nicely written.

This reviewer has very minor suggestions.

Figure 1: The word "PS" is not expanded. In addition, authors have not discussed conserved sequence elements in SaRNA... what about the 5' and 3' UTR sequences? or some specific sequences embedded in the SaRNA sequence that play their role?

Figure 2 is redundant, there is no vital information happening through this figure. Rather, I would think of preparing another possible reasonable figure.

VoCs---- please expand at first instance.

It would be interesting to the audience to elaborate more on the dose-limiting reactogenicity among the different RNA delivery systems.

Lines 395-397: Could you please elaborate on the mechanism of release of SaRNA from DREP in transfected cells?

Lines 523-526: An appropriate reference should be cited.

Author Response

Dear Editor and Reviewers,

The review manuscript entitled “Self-amplifying RNA: a second revolution of mRNA vaccines against Covid-19” has been revised according to the Reviewer´s suggestions. Please find below a point-by-point reply to each of your comments. All the new data and changes suggested by you have also been highlighted in red in the manuscript.

Reviewer 2:

Figure 1: The word "PS" is not expanded.

PS has been expanded in the Figure legend (packaging signal)

In addition, authors have not discussed conserved sequence elements in SaRNA... what about the 5' and 3' UTR sequences? or some specific sequences embedded in the SaRNA sequence that play their role?

We have added in Figure 1 the conserved sequence elements that were missing, such as the 5´and 3´untranslated regions (UTR). A short sentence explaing these elements has also been added in the main text (lines 138-139). The packaging signal (PS) has also been expanded as requested by the reviewer in the previous point.

Figure 2 is redundant, there is no vital information happening through this figure. Rather, I would think of preparing another possible reasonable figure.

A new Figure 2 has been prepared where more vital information has been included

VoCs---- please expand at first instance.

It was already defined at first instance in line 38 (first paragraph of the manuscript).

It would be interesting to the audience to elaborate more on the dose-limiting reactogenicity among the different RNA delivery systems.

Following the reviewer´s suggestion we have included more information regarding the dose-dependent reactogenicity of saRNA vaccines observed in clinical trials (lines 781-790).

Lines 395-397: Could you please elaborate on the mechanism of release of SaRNA from DREP in transfected cells?

We have now included a brief explanation of the DREP platform (lines 402-404).

Lines 523-526: An appropriate reference should be cited.

An appropriate reference has been added (ref 84 in line 539).

Reviewer 3 Report

Comments and Suggestions for Authors

The manuscript by Silva-Pilipich et al discusses the introduction of vaccine development based on the self-amplifying RNA (saRNA). The review discusses SARS-CoV-2 virus, COVID19 disease and vaccines, saRNA and delivery strategies to increase efficacy of saRNA based vaccines. The review is too long and full of unnecessary details which makes it uninteresting. Authors should consider making original and innovative figures, graphs and tables, which will benefit a reader. I would not suggest that the review should be considered in current form. 

I have provided my comments to the authors in the following section.

Author Response

Dear Reviewer,

The review manuscript entitled “Self-amplifying RNA: a second revolution of mRNA vaccines against Covid-19” has been revised according to the Reviewer´s suggestions. Please find below a point-by-point reply to each of your comments. All the new data and changes suggested by you have also been highlighted in red in the manuscript.

Reviewer 3:

The manuscript by Silva-Pilipich et al discuss the introduction of vaccine development based on the self-amplifying RNA (saRNA). The review discusses about SARS-CoV-2 virus, COVID19 disease and vaccines, saRNA and delivery strategies to increase efficacy of saRNA based vaccines. The review has been written nicely, however; I have few suggestions here for the authors.

  1. The review is too long and very repetitive. It lacks relevant figures and even current figures in the manuscript are not very appealing.

Authors should find innovative way to present their massive data-based review to more attracting to readers. Authors can use figures, graphs or tables to compile the review in a shorter form. There are many places where irrelevant and generic sentences have been written, which are occupying precious space and making the manuscript unnecessary longer. See line 328-331.

The review has been shortened by removing unnecessary details to make it more attractive. We have also made substantial changes in some figures (i.e. see new Figure 2). Tables have also been revised and improved by adding missing information.

  1. I will suggest authors to prepare a table, providing the details of major approved COVID19 vaccines from various countries and what plateform did they use in developing vaccines. In 2024, it is not justifiable to discuss just 2 or 3 vaccines in the review. See line 70-84.

I new table (Table 1) with details of the most relevant approved COVID19 vaccines from various countries has been added.

  1. Line 46, is it really “metastable”? There are several crystal and cryoEM structures of spike protein alone in postfusion conformation, please explain.

We have changed the word “metastable” by “stable”, which migh reflect better the conformation of the prefusion spike protein (line 46)

  1. Please highlight of shortcoming of nucleic acid-based vaccines, specifically mRNA based.

A sentence has been added to highlight some of the shortcomings of mRNA-based vaccines (lines 110-112): “Despite the good performance of mRNA vaccines they also present some shortcomings, such as their high cost and the fact that their storage and transportation require very low temperatures, rendering them inaccessible for many low-income countries.”

  1. Provide references for each point, line 137-141.

New references have been added for each point (refs: 24-27 cited in lines 150-153)

  1. Briefly describe in the manuscript that saRNA induce interferon responses with suitable references.

This is described in line 153, where two new references have been included (26-27)

  1. Line 166-174, it is well known and too general. It can be left out to gain some space.

This paragraph has been shortened following the reviewer´s suggestion. However, we did not remove it completely since we believe that it is important to clarify which animal models are suitable for testing COVID-19 vaccines

  1. Line 210-215. Explain, how wuhan isolate antigen could elicited antibodies that could neutralize Alpha, Beta and Delta variants?

The following sentence has been added in line 227-228: “This crossreactivity is likely due to the fact that most VoCs share multiple epitopes, differing only in a few amino acids.”

  1. Line 221, Which SARS-CoV-2 strain was used for challenging intranasally?

Hamsters were challenged with the USA-WA1/2020 virus isolate, which corresponds to the original Wuhan strain (added now in line 234)

Round 2

Reviewer 3 Report

Comments and Suggestions for Authors

The review manuscript by Silva-Pilipich et al discusses the self-amplifying mRNA vaccine development against COVID19. The revised manuscript is better than the previous version. However, I have the following minor comments.

1.     Line 46, it is not really the case that all virions contain 100 spike proteins, and all 100 spike proteins are in a stable prefusion conformation until they bind to a receptor. Please correct this and try to be very precise in your sentence. If you think that your sentence is scientifically correct, then provide a few genuine references. 

2.     Line 114, I will strongly suggest removing “many low-income countries” wording from the sentence. It is a scientific paper, not a NEWS article, therefore, it would be more apt to use scientific words.

3.     Authors should work to make Figure 1 better. The Figure and its legends do not correlate very well. 

4.     Line 226-229. In review phase 1, I had raised a concern in the comment number 8. I am not satisfied with the added sentence. It means that the sera contains one or more broadly neutralizing antibodies. Authors should provide details of such broadly neutralizing antibodies, if this is the case. Discovery of such antibodies is the key in fighting with COVID19. In case your reference does not have a proper experimental proof then avoid including such studies in your review.

5.     The review manuscript is still very long and the last 13-14 pages have no figure related to the text.

Comments on the Quality of English Language

It has areas which can be improved. 

Author Response

Dear Reviewer,

The review manuscript entitled “Self-amplifying RNA: a second revolution of mRNA vaccines against Covid-19” has been revised according to the Reviewer´s suggestions. Please find below a point-by-point reply to each of your comments. All the new data and changes suggested by you have also been highlighted in red in the manuscript.

Reviewer 3:

The review manuscript by Silva-Pilipich et al discusses the self-amplifying mRNA vaccine development against COVID19. The revised manuscript is better than the previous version. However, I have the following minor comments.

  1. Line 46, it is not really the case that all virions contain 100 spike proteins, and all 100 spike proteins are in a stable prefusion conformation until they bind to a receptor. Please correct this and try to be very precise in your sentence. If you think that your sentence is scientifically correct, then provide a few genuine references.

We have now slightly changed the sentence indicated by the reviewer, not mentioning the number of spike trimers on the surface of the virion, since this information is not relevant for this review. In addition, we also wrote that “the spike is usually in a stable prefusion conformation”, leaving it open that not all spikes might be in this conformation (line 46).

  1. Line 114, I will strongly suggest removing “many low-income countries” wording from the sentence. It is a scientific paper, not a NEWS article, therefore, it would be more apt to use scientific words.

According the reviewer´s suggestion we removed the words “low income countries” (line 113).

  1. Authors should work to make Figure 1 better. The Figure and its legends do not correlate very well.

We have now improved the figure legend to correlate better with the image.

  1. Line 226-229. In review phase 1, I had raised a concern in the comment number 8. I am not satisfied with the added sentence. It means that the sera contains one or more broadly neutralizing antibodies. Authors should provide details of such broadly neutralizing antibodies, if this is the case. Discovery of such antibodies is the key in fighting with COVID19. In case your reference does not have a proper experimental proof then avoid including such studies in your review.

We thank the reviewer for this observation. However, it has been shown that individuals immunized with the orginal Pfizer–BioNTech BNT162b2 vaccine (from Wuhan strain) when infected by Omicron have substantial, but not complete, escape from vaccine-induced neutralizing antibodies, suggesting that at least some neutralizing antibodies could be present, although they might no be very abundant or might not have an optimal activity (Cele et al. Omicron extensively but incompletely escapes Pfizer BNT162b2 neutralization. Nature 2022 Feb;602(7898):654-656). We have now rewritten the sentence in lines 235-243 emphasizing the need to induce broadly neutralizing antibodies.

  1. The review manuscript is still very long and the last 13-14 pages have no figure related to the text.

We have now reduced the length of the manuscript. We believe that since both the new figure 2 and tables 2 and 3 summarize the most relevant information present in the last part of the text and adding a new figure in this part could be a bit redundant.
